# Cost of Water Use for Negotiating Rates in Energy Exchanges: Evidence from the Hydroelectric Industry

**Jair-Albeiro Osorio-Agudelo [1,3,*], David Naranjo-Gil [2] and Vicente Ripoll-Feliu [3]**

[1]   Department of Accounting Sciences, Faculty of Economic Sciences, Universidad de Antioquia, 050010 Medellín, Colombia
[2]   Departamento de Economía Financiera y Contabilidad, Facultad de Ciencias Empresariales, Universidad Pablo de Olavide, 41013 Sevilla, Spain; dnargil@upo.es
[3]   Accounting Department, Universidad de Valencia, Faculty of Economics, 46070 Valencia, Spain; vicente.ripoll@uv.es
*   Correspondence: albeiro.osorio@udea.edu.co; Tel.: +54-312-286-00-79

**Abstract:** This paper analyzes the importance of the cost of dam water use in hydroelectric generators according to the International Financial Reporting Standard (IFRS) and Management Accounting. Not valuing the use of water from dams would imply undervaluing energy generation service, leading to a lack of reasonability in the Financial Statements of electricity generators. For this reason, it is vital to recognize that dam water has a cost that directly impacts the Statement of Financial Position as an asset and later, in energy generation and commercialization, its cost will impact the statement of profit or loss, according to the IFRS as good accounting and financial practices around the world. Said cost will also be an important indicator for rationalizing consumption, defining public policy, or determining energy tariffs. An empirical study is conducted for Colombia and Norway, two of the main countries in the world whose primary source of energy generation is water. The results evidence the need for hydroelectric generators to present the cost of their hydric reserve as intangibles inventories because of its potential capacity to generate electric power. Additionally, there is a positive and significant relationship between the cost of water and the price of energy, and a negative relationship between the price of energy and dam levels.

**Keywords:** water cost; hydric reserve; hydraulic generation; IFRS; activity-based costing; dam; services inventory; energy prices; management accounting

---

## 1. Introduction

Water for human, government, and society use is consolidated as the main concern of the 21st century and can become a great determinant of countries' wealth and quality of life; however, economic development is gradually reducing its availability [1,2]. For this reason, it is necessary to assign a cost to water to help understand that this is a valuable resource. Very few models of hydraulic energy price determination are centered on the cost of dam water use. Nor have these models considered other values of environmental, social, and economic nature; impossible to disregard today [3].

Water and its related policies are complex in themselves and are linked to domains fundamental to development, such as the environment, agriculture, energy, and health [4,5]. Evidence shows that there does not exist a single solution for the problems of water scarcity; it rather varies according to the geographical position and thus it is important to adapt the policies to each country's conditions [4].

To tackle current and future challenges, the water governance of the Organization for Economic Co-operation and Development (OECD) helps in the designing and implementation of solid public

policies allowing to harvest economic, social, and environmental benefits from good water governance, in which one of the principles is based on ensuring that the governance frameworks help mobilize water finances and allocate financial resources in an efficient, timely, and transparent manner, from solid budget-planning and accounting practices that show a clear image of water-related activities and any other situation associated with it [4].

The main source of renewable energy generation is hydroelectric energy, which represents 71% of the total renewable energy production and 16.4% of the world energy for 2016 [6]. However, countries such as China, the United States, and Russia present a lower participation of hydroelectric energy over other energy-producing sources (coal, natural gas, oil, nuclear power, etc.), whereas in countries such as Norway, Colombia, Paraguay, Canada, Brazil, and Venezuela this participation is much higher (See Appendix A, Table A1) [7,8].

Norway is ranked 14th among the 20 main countries in the world with unutilized hydroelectric potential with 161,000 Gigawatt-hour GWh/year, while Colombia comes in 16th place with 151,000 Gigawatt-hour GWh/year (See Appendix A, Table A2) [6]. On the other hand, access to information is feasible and of primary source, and Norway is a reference in hydraulic energy production in Europe, as is Colombia in Latin America in topics related to energy regulation and policies of public services management in general. For these reasons, the empirical case is developed with data from Norway and Colombia.

For the above, the main aim of this work is to analyzes the importance of the cost of dam water use in hydroelectric generators according to the International Financial Reporting Standard (IFRS) and Management Accounting. To achieve this aim, econometric estimations are performed to model the variation in energy price on the Energy Exchange and a variable indicative of the cost of dam water use for both countries, and thus demonstrate the influence of water inventory on energy prices as well as on the Statement of Financial Position and Income Statement.

In this way, it can be recognized that dam water has a cost that is a determining factor in defining public policy and determining tariffs, which could help hydroelectric generators around the world to implement methodologies or strategies to determine its costs, saving, or rationalization and the adequate presentation of reports in their financial statements [6,7]. Potential investors could use it to decide on the possibility to invest or not in hydroelectric companies; and even the government can find specific information to generate its public policy.

The remainder of the work is structured as follows: Section 2 presents the literature review and explains the cost of dam water use in hydroelectric generation, Section 3 addresses the material and methods, Section 4 shows the results, and the final section presents the main findings and conclusions.

## 2. Literature Review

Accounting is known as the financial language of businesses, through which economic information is communicated to the people who have a stake in an organization: managers, shareholders and potential investors, employers, creditors, and the government [9]. However, accounting is a changing phenomenon that is continuously redefined and converges to new realities [10]. The International Financial Reporting Standards (IFRS), issued by the International Accounting Standard Board (IASB), headquartered in London, are currently applied by most countries around the world. IASB's main aim is to develop a unique set of high-quality financial reporting standards that is understandable, required, and accepted globally, which contribute to the preparation of financial statements and which are useful for stakeholder decision-making [11].

Companies must proceed according to the dynamics of globalization and economic internationalization that have set a significant pace on the international regulation process; it is thus how international regulation in accounting matters, since the appearance of the International Accounting Standards Committee (IASC) is made stricter with the permanent adjustments and transformations to the International Financial Architecture [12,13]. In this way, in the area of international accounting regulation a process of consolidation of supranational bodies is experienced. Today the International

Accounting Standard Board (IASB, formerly IASC) stands as the "most accepted" body that issues regulation of business organizations' financial information [12].

Thus, "convergence" projects intending to approximate the developments of the IASB with national regulating institutions such as the Financial Accounting Standard Board (FASB) of the United States, generate strong interests in the international accounting regulation process [12]. The regulation approach has moved from the regulation of practices for the production of financial accounting to the setting of principles for the production of financial reports, typical of the North American regulating tradition, and is strongly related with the structural conditions of the world's most developed and biggest financial and stock market [12].

The benefits of implementing IFRS include harmonization of accounting practice in adopting countries which, in turn, leads to greater comparability, lower transaction costs, and an improvement in international investment [14]. Essentially, IFRS adoption gives a positive signal of greater transparency and quality of accounting information [15]. Companies that voluntarily or compulsorily implement IFRS tend to disclose higher quality financial statements than those that use their nationally adopted Generally Accepted Accounting Principles (GAAP) [16,17].

Application of IFRS by hydroelectric generation companies contribute to the quality of their financial information; if the IASB continues improving the quality of IFRS, it would be wise to expect financial reports to be more relevant and reliable each time [18]. Additionally, the main benefits derived from IFRS adoption are transparency and comparability of financial statements and a lower degree of opacity in comparison to national standards, which will lead to an increase in foreign direct investment [19–21]; undoubtedly, their application has consequences for investors by giving more credibility to the prediction capacity of the figures [22].

The hydric reserve used for energy generation should be valued according to the International Accounting Standard (IAS) No. 2, which establishes that inventories are tangible or intangible assets (a) owned to be sold in the normal course of operation, (b) in production process to be sold later, or (c) in the form of materials or supplies that will be consumed in the production process or in the provision of services [23]. One of the main innovations of IAS 2 for hydroelectric generators is the possibility to present hydric reserve as inventories in their Financial Statements, thereby allowing to capitalize labor cost and other attributable costs incurred in the energy production process [24]. Said capitalization is initially presented as higher hydric reserve value (intangible inventory) in the Statement of Financial Position and later at the time of selling as the cost of energy sold in the statement of profit or loss [23,25].

Management Accounting is an important complement to IFRS to determine in a more detailed manner the cost of hydric reserve in dams, since relevant financial and non-financial information is given for decision-making, the right allocation of resources and the monitoring and evaluation of performance. Some examples of information obtained with management accounting are the cost of producing a product, the cost of delivering a service, and the cost of carrying out a business activity or process [26]. One of the main aims is to improve the quality of the information generated on the cost of products and services [27].

Modern developments in Management Accounting began with the introduction of activity-based costing (ABC) in the early 80s, which identifies activities within an organization and assigns the cost of each activity with resources to all products and services according to the actual consumption of each activity [28,29]. Implementing ABC brings great benefits such as increasing operating profits by assigning general costs based on the consumption of resources for each activity; allowing to visualize where the most important costs are generated and what originates them; providing growth by eliminating the bottleneck that was causing a capacity restriction; and possibly helping in administrative and strategic decision-making [27,30].

ABC fosters a better understanding of the operations, is relatively easy to use, and produces a great variety of accessible information, the true power of such system being its capacity to promote and

enhance understanding of the management of the activities and processes and contribute to improving their management [31,32].

Financial accounting and management accounting are going through a drastic development stage. Therefore, paying attention to the convergence phenomenon may contribute to potentially more significant results and to more practical and relevant theories in accounting research [10].

### 2.1. Cost of Dam Water Use in Hydroelectric Generation

Multiple categorizations of hydroelectric plants can be found in the literature review, such as the administrative classification where they are divided according to their power and technical classification, mainly regarding their way of operation. This work takes the hydroelectric plants classification from the technical perspective, which divides them into four major groups: dam plants, run-of-river plants, mixed plants, and others (microplants and plants in drinking water circuits) [33]. The main interest is focused on dam plants that store water and have enough capacity to enable regulation of the maximum flow in one day; this regulation capacity is used to deliver energy at times of high demand, yielding two benefits: It permits to regulate the energy market; and the sale of energy can be made when it reaches high prices [33,34]. Additionally, this type of dams have functions that bring benefits such as regulating the flow of water to avoid flooding, supplying water for agriculture and industrial purposes, among others [33].

The costs for decision-making is one of the most important characteristics of management accounting in relation to other indicators [35]. The importance of the cost of water use is undeniable for accounting, financial, and decision-making purposes, such as price calculation.

One alternative to value the cost of dam water for energy generation is to identify the high initial investment which presents long building periods and very low operating costs, divided into pre operating and operating costs [36,37]. Among the former are the investment costs, which are significantly determined by the specific location conditions of each project (land; infrastructure; civil works; hydro mechanical generation equipment and ancillaries; environmental investments; engineering; and construction, equipment, financial and legal unforeseen events) [36,37]; these costs must be depreciated along the dam's life cycle. Operation costs correspond to the functioning costs of the generation company (costs of maintaining power lines, roads, etc., besides the costs of environmental management, insurance and legal charges) [36,37].

To keep the life cycle of a dam and ensure an efficient energy generation process, different activities must be taken into account, such as erosion and sediment control, which consists in a previous study of the basins with the biggest erosion [36,37]. Table 1 displays in more detail each one of the cost concepts of a dam for hydroelectric generation [36,37].

In general terms, hydroelectric generating companies have different alternatives to calculate the cost of water use, which they can use for multiple purposes. From the financial point of view, the cost of water use for hydroelectric generation can be calculated in terms of fixed and variable cost elements; therefore, there are different cost definitions on which water valuation is based [15].

Different management accounting methodologies can be used to assign a value to water. Water use cost refers to several components such as cost of capital, operation cost, maintenance, and administration cost, the reliability of the supply cost in terms of quality and amount, opportunity cost, and the cost of externalities imposed for its exploitation [12].

According to the IFRS issued by the IASB and the IPSAS stemming from the International Public Sector Accounting Standards Board (IPSASB), both the International Accounting Standard No. 2 (IAS 2) and the International Public Sector Accounting Standards No. 12 (IPSAS 12), energy must be measured and inventoried, besides including as inventories the strategic energy reserves for emergency cases or other situations [23,25].

Subsequently, the resources should be assigned to the energy generation service (the object of costing) through a cost model. One of the most commonly used models is activity-based costing (ABC), whose important contributions are the accuracy improvement in the cost calculation of any cost target,

as well as the determination of its profitability, by avoiding possible cross-subsidies between products or services, the outputs of said system being superior to those obtained through conventional costing systems [31,32,38,39].

**Table 1.** Costs concepts of a hydroelectric plant dam.

| Cost Concept | Definition |
| --- | --- |
| Land | The aliquot of the land value of the plant or of the servitudes required during all its life cycle. |
| Infrastructure | The aliquot of the access roads, connection lines, and campsites and offices during its life cycle. |
| Civil works | The aliquot of the costs of the physical infrastructure required for exploiting the hydric resource, such as dam works, abstractions, desanders, power lines, discharge, powerhouses, and substations during its life cycle. |
| Hydro mechanical equipment of generation and ancillaries | The aliquot of turbines, generators, control equipment, gates, drains, valves, fire systems, air conditioning, oil and water pump, diesel plants, transformers, among others, during its life cycle. |
| Environmental investments | The aliquot of the investments and previous studies in the environmental area, without including management plans. |
| Engineering | Costs of technical and environmental management, design, technical auditing during the project construction. |
| Unforeseen events (preoperational) | Construction and equipment unforeseen events. |
| Financial (preoperational) | Given by costs escalation during the construction period and preoperational interests. |
| Legal (preoperational) | Legal charges that may apply during the construction period. |
| Operation and maintenance (OM) Fixed component | Costs of the generation company and functioning and maintenance costs of works and equipment |
| Environmental management | As a percentage of total investments. |
| Insurance | Cost of insurance the project must assume annually for the normal risk coverage. |
| Operating legal charges | Applicable legal charges during the operation of the project, depending on each technology, type of plant and region. |
| Maintenance costs | Erosion and sediment control, reservoir bank protection system and waste control and management. |

ABC calculates the cost as the sum of all inputs plus the costs of each one of the processes and activities carried out within the company and is founded on two premises: products or services require processes and activities and these processes and activities are the ones that consume resources that generate cost. To properly apply this methodology in hydroelectric companies, the main thing to do is to identify the relationship between the resources consumed in the processes and the reason why the cost is caused [31,38,39]. Table 2 shows the processes and activities in the generation of hydraulic energy [31].

**Table 2.** Processes and activities of the hydraulic generation business.

| Process | Activities |
| --- | --- |
| Hydraulic energy generation | Planning<br>Operation |
| Maintenance | Coordinating and managing maintenance<br>Electric preventive<br>Electric corrective<br>Mechanical preventive<br>Mechanical corrective<br>Civil preventive<br>Civil corrective |
| Environmental management | Management of hydrographic basins<br>Environmental monitoring<br>Environmental licenses |
| Energy commercialization | Market analysis<br>Plant offer<br>Energy purchase<br>Energy sale |
| Customer attention | Information supply<br>Service to third parties |
| Billing and collection | Measurement<br>Liquidation of energy transactions<br>Billing<br>Collection |
| Energy contracts management | Contracts management |

The total cost of hydraulic energy generation is the sum of the costs of each one of the processes and activities necessary for generating energy and delivering it to the different clients and users of the service, as follows:

$$CEG = CHEGP + CMP + CEMP + CECP + CCAP + CBCP + CECMP \tag{1}$$

where,

CEG: cost of energy generation
CHEGP: cost of hydraulic energy generation process
CMP: cost of maintenance process
CEMP: cost of environmental management process
CECP: cost of energy commercialization process
CCAP: cost of customer attention process
CBCP: cost of billing and collection process
CECMP: cost of energy contracts management process

IFRS determine that the cost of dam water used for hydroelectric generation must be measured taking into account the energy acquisition costs. These are the transformation costs comprising the costs directly related to the energy generated, such as the direct personnel cost and a systematically calculated part of the indirect, variable, and fixed costs incurred for transforming water into energy; and the rest of the costs providing they were incurred in the process of giving energy its current condition and location [23].

In this sense, of the costs detailed in Equation (1) for accounting measurement purposes according to the IFRS, the three first processes must be considered as inventoriable costs, which are presented in the current assets of the Statement of Financial Position under the denomination of potential energy inventory, which represents dam water cost use. The next four processes of Equation (1) are considered non-inventoriable costs and must be presented directly in the income statement.

## 2.2. Energy Structure in Norway and Colombia

In Norway, energy production by technology, as percentage of the total for the year 2018, has a participation of 95.02% in hydroelectric power plants, followed by wind with 2.63%, and natural gas with 1.77%, among others. The lowest participation is nuclear, geothermal, solar, and tidal with 0% [40].

The electricity market in the Nordic countries (Denmark, Finland, Norway, and Sweden) was deregulated in 1991 and is considered a mature and liquid market so it is a model to be followed by many countries; and it is to a great extent harmonized with the European Union legislation [41,42]. Production over the course of a year is determined by all the factors affecting the electricity market prices such as household demand, hydrological balance, reserve levels, and the market conditions both in the Nordic countries as in the north of Europe [43].

In Norway, a hydroelectric plant connected to a dam must use the available water or store it to later sell electricity at a higher price [43]. Wholesale production and prices are normally higher in winter, mainly because of the high residential demand [43].

In the case of Colombia, water represents a competitive advantage, as well as for hydroelectric companies, for being part of the main determining factor of competitiveness according to Porter [44], given it is the second hydrologically richest country (rivers, Andean highlands, wetlands, basins, etc.,) in Latin America according to the 2014 statistics on renewable internal freshwater resources published by the World Bank [45], and for being surrounded by two oceans. In spite of this, efficiency in the utilization of water must be promoted since the hydrologic richness is found in relatively unpopulated areas such as the Amazonia and the Orinoquia, while in urban and industrial zones there is scarcity of this resource, which is the basis of hydroelectric generation, as stated in Law 697 of 2001 [46,47].

The production by technology in Colombia for the year 2018 was 82.17% in hydroelectric plants, followed by thermal power plants (gas, coal and fuel) with 16.69%, biomass with 1.06%, wind with 0.06%, and solar with 0.02% [48,49].

In Colombia energy deregulation began in 1994, and the spot market started operations in July 1995, supported by Laws 142 and 143 (Congreso de la República de Colombia, 1994) [50,51]. The Colombian model was adapted from the English model, although its situation is similar in many aspects to the situation of Norway before the creation of the Nord Pool, with respect to generation technology, mainly because of dependence on hydraulic energy [52]. This new reform introduced competition, established a new industrial structure and a new independent regulatory institution called CREG (Comisión de Regulación de Energía y Gas—Energy and Gas Regulation Comission), set the bases for the expansion and diversification of electric generation sources and improved efficiency and reliability of the sector [53–55]. The Colombian electricity market system is price-based, where firms submit daily bids of both energy and prices to the Centro Nacional de Despacho (CND), which is the official system operator at the national level [51].

In this country, energy exchange prices fluctuate according to supply and demand, which allows generators to establish strategies through mathematical models such as the hydrothermal dispatch, which allocates the optimal generation to each plant, minimizing the operating cost in each time period, with the aim of inducing a price increase and improving profitability, regardless of the efficient utilization of the hydric resource [56,57].

In a study of the market fundamentals affecting the formation of electric energy prices in Colombia, some variables were found which cause a positive effect on price formation, such as demand, hydrology, and declared availability [58]. In periods of strong droughts, water acquires greater importance than in winter periods. A country that depends on hydroelectric generation must migrate toward increasing thermoelectric production and urge all the firms to produce their own energy. It is thus how the government, in the regulated market, authorizes to increase energy tariffs in household consumption in an effort to achieve efficient consumption and, in the unregulated market, demand is greater to supply, which brought a drastic tariff increase [59]. The regulated market refers to the natural or legal person whose electricity purchase is subjected to the tariffs established by the CREG, whereas the unregulated market is the natural or legal person that has an energy demand above 2 Megawatts MW [50].

## 3. Materials and Methods

This research presents the correlation between the hydric reserve in the dams and the price of energy (as dependent variable), and per capita energy consumption (as explanatory variable), controlling the effects by the presence or absence of El Niño phenomenon; with which generators transfer their valuation of the service to the market; resorting to models that help evidence the relationship between said variables and demonstrating the importance of the cost of water use in hydraulic generation. The hydric reserve variable will help identify whether these companies are imputing a cost for the use of water in the supply price.

The function of the econometric model considered the available monthly historical data from January 2000 through December 2016 of all the variables for Colombia and Norway, which results in a considerable sample. The data series for Colombia are taken from the primary source, that is, the operator of the Mercado Energético Colombiano (MEM), the World Bank and the Compañía Expertos en Mercado (XM), which is a state industrial and commercial company specialized in systems management in real time in the areas of energy, finances, and transport [56,60]. For Norway, information is taken from the data base of the Nord Pool, which is the largest energy commercializing entity in Europe, the International Energy Agency, the World Bank, and The Norwegian Water Resources and Energy Directorate (NVE) [43,60–62]. We used Stata SE (programming language for statistical analysis) to obtain the results. The variables used in the study were the following:

- Price of the National Exchange (Elspot Price): Corresponds to the supply price in $/kWh and in EUR/kWh of the last flexible plant to cover the national commercial demand, plus the increase to

remunerate the costs not covered of thermal plants in the ideal dispatch [56]. Dependent variable. For the Colombian case, this variable is available daily so it was averaged to obtain a monthly data; for Norway, the data are available monthly by geographic area, so they were averaged obtaining a total monthly data for the country.

- Hydric Reserve: Corresponds to the volume of the dams (kWh). Potential Energy Inventory [56]. Explanatory variable. These data are available daily per region for Colombia, so the regions were added up and then the monthly average was calculated for the country. For Norway, the data are available on a daily basis, so they were averaged to obtain a monthly data.

- Per capita energy consumption: Average value of consumption per inhabitant estimated for a given moment. The measurement unit is kilowatts hour (kWh). Explanatory variable. The annual variable was obtained for both countries, so it was kept as a constant within the months observed for each year.

- El Niño phenomenon: Temperature warming that implies noticeable changes in the sea surface and the atmosphere in the equatorial Pacific Ocean [63]. Dummy variable used only for the Colombia model.

- Seasons of the year: Represents the periods of the year in which climatic conditions stay within a certain range. These periods are normally four, they last three months approximately and are spring, summer, autumn, and winter. Dummy variable used only for the Norway model.

This correlation between energy price and hydric reserve is estimated in the form of a long-term relationship or level relationship, recognizing the short-term dynamics implicit in this type of relationships. The characteristics of the time series data for the estimation are verified through unit root tests to confirm their seasonality, applying the Augmented Dickey-Fuller (ADF) test (See Appendix A, Table A3). A total of 12 lags is included for all the variables in the test. Tests on price and consumption include trend, and the test on the hydric reserve includes derive only.

The unit root test points that both energy price and per capita consumption have unit root, while hydric reserve is zero-order integrated. This mix of seasonal and non-seasonal variables implies that the long-term relationship between the variables cannot be estimated via OLS (Ordinary Least Squares), nor can the presence of cointegration between them be directly verified. For this, the autoregressive distributed lags (ARDL) model methodology is used, verifying whether or not there is a long-term relationship between the variables through the Bound Test of Pesaran, Shin and Smith (PSS) [64]. Specifically, we will estimate a model of the type:

$$y_t = a + A(L)y_t + D(L)x_t + \varepsilon_t, \tag{2}$$

Where $a$ is the intercept, $A(L)$ represents an autoregressive polynomial, $y_t$ is the variable of interest, $D(L)$ is the polynomial gathering the lags of the explanatory variables $x_t$, and $\varepsilon_t$ is the error term. Once the existence of a long-term relationship (level relationship) of the variables is determined, this model is re-written in the form of an error correction equation as:

$$\Delta y_t = b + \sum \alpha_i \Delta y_{t-i} + \sum \beta_j x_{t-j} + \phi z_{t-1} + \epsilon_t, \tag{3}$$

Where $b$ is the intercept, $\alpha_i$ are the coefficients of each of the lags of $y_{t-i}$ taken into account in the model, $\beta_j$ are the coefficients of each of the lags of the variable $x_{t-j}$, $\phi$ represents the error correction parameter and $z_{t-1}$ represents the long-term equilibrium deviations and $\epsilon_t$, is the error term. For the Colombian variables, we work with the ARDL (3,12,3) model, that is, a model including three lags of the dependent variable, twelve lags of the hydric reserve logarithm, and three lags of the per capita energy consumption logarithm. With this parameterization and including the constant, the PSS is conducted. For Norway, the seasonality of the variables is tested through the ADF and PSS test to verify whether there exists a long-term relationship between the variables and the estimation via OLS.

For the joint model, since the temporal dimension is greater than the cross-sectional one, we work with a time series scheme: conducting unit root tests and concluding from them the best methodology to implement in the process of estimation of the relationship between the interest variables. The tests to be applied on the panel variables are the Levin, Lin, and Chu (LLC) and the Breitung tests, of standard use in these cases [65,66].

## 4. Results

### 4.1. Colombian Case Model Results

When carrying out the Bound Test, it is identified that with both the F test and the t test, the critical value is above the interval's upper limit, regardless of the significance level, so the null hypothesis of no level relationship is rejected, and the next step is to establish the error correction equation for the variables involved in the estimation (See Appendix A, Table A4).

The results of the error correction model (See Table 3) show the following: (1) The error correction term is equal to −0.210, significant at all standard significance levels; its negative sign indicates that, in effect, all long-term equilibrium deviations are temporal, which means there is a level relationship of the variables. (2) The long-term relationship indicates that increases of 1% in hydric reserve are associated with reductions of 0.155% in energy price, although this coefficient is not significant. (3) Increases of 1% in per capita energy consumption translate into increases of 2.389% in energy price, this result being significant at any standard significance level. (4) The fact that El Niño phenomenon occurs causes the energy price to have a temporary increase of 0.197%; this effect is part of the model's short-term dynamics. The complete results can be seen in Appendix A, Table A5.

**Table 3.** Error correction model.

| Variables | Adjustment (ADJ) | Long Run (LR) | Sho8rt Run (SR) |
|---|---|---|---|
| El Niño | - | - | 0.197 ** |
| - | - | - | (0.088) [1] |
| Ln (Hydric reserve) = L, | | −0.155 | |
| | | (0.947) | |
| Ln (Per capita energy consumption) = L, | | 2.389 *** | |
| | | (0.413) | |
| Ln (Exchange Energy Price) = L, | −0.210 *** | | |
| | (0.048) | | |
| Constant | | | 0.068 |
| | | | (3.230) |
| Observations | 192 | 192 | 192 |
| R-squared | 0.510 | 0.510 | 0.510 |

Notes: [1] Standard errors in parentheses. *** $p < 0.01$, ** $p < 0.05$

To verify whether the short-term dynamics were well captured in the model, tests on the residuals are carried out: unit root and no autocorrelation (See Appendix A, Table A6). The test implies that the null hypothesis of unit root of the residuals is rejected at any standard significance level. These are seasonal. On the other hand, the no autocorrelation Portmanteau test is performed for lags 1, 6, and 12. In no case the null hypothesis of no autocorrelation is rejected, indicating a good behavior of the residuals, and hence, a good model specification (See Appendix A, Table A7).

### 4.2. Norwegian Case Model Results

The unit root test results show that the null hypothesis of no seasonality is rejected in the cases of energy price and hydric reserve, meanwhile it is concluded that per capita consumption is non-seasonal (See Appendix A, Table A8). The unit root test was conducted with constant for energy exchange price and hydric reserve, and with trend for per capita consumption. Once again, we have a mix of seasonal

and non-seasonal variables, so the PSS test is applied to verify whether there is a long-term relationship between the variables (See Appendix A, Table A9).

In this case, the test cannot be rejected at 1% so we conclude that there is no long-term relationship and the regressors are zero-order integrated. Given this, the level relationship we seek to obtain can be estimated via OLS, having the precaution of carrying out the estimation with robust errors because of the presence of autocorrelation and heteroscedasticity, so that the inference be reliable. In this sense, the estimation is performed via OLS with robust errors using the Newey-West estimator (See Table 4). The results indicate that: (1) Increases of 1% in hydric reserve in Norway are associated with falls of 1.041% in energy price in said country; (2) changes in consumption do not have a significant effect on energy price; and (3) seasonal dummies show a significant effect of spring and autumn, the effect being positive in the case of autumn and negative in the case of spring.

**Table 4.** Regression model for the Norway case.

| Variables | Ln (Exchange Energy Price) |
|---|---|
| - | - |
| Ln (Hydric reserve) | −1.041 *** |
| - | (0.161) [1] |
| Ln (Per capita energy consumption) | −0.762 |
| - | (1.079) |
| Winter | 0.077 |
| - | (0.105) |
| Autumn | 0.328 *** |
| | (0.099) |
| Spring | −0.569 *** |
| | (0.135) |
| Constant | 24.311 *** |
| | (4.727) |
| Observations | 204 |

Notes: [1] Standard errors in parentheses. *** $p < 0.01$.

### 4.3. Joint Model's Results

The information on both countries is gathered in a data panel with a higher number of freedom degrees and thus the estimation results are more robust (See Appendix A, Table A10).

The results of the panel unit root tests do not allow ruling out the presence of unit root in the panel variables, so the next step is to carry out a cointegration panel test. The test to be applied is that of Westerlund, based on error correction, which consists of testing whether there is a representation of error correction between the variables involved in the regression, and to do so panel tests are derived that judge the cointegration hypothesis for the panel as a whole [67]. With this test, the null hypothesis is that there is no error correction against an alternative hypothesis that there is error correction.

Therefore, the rejection of the null hypothesis is taken as cointegration evidence. Although it is not probable in the present exercise, the p-values of the test will be calculated through the bootstrap with 1000 repetitions to obtain robust results to cross dependence. In this test the short-term dynamics are captured through the inclusion of leads and lags of the variables in differences, and the order of these leads and lags is determined through the information criteria (See Appendix A, Table A11).

The p-values of the sample imply that the null hypothesis of no error correction should be rejected at a significance level of 5%. Hence, we conclude that the variables are cointegrated, as occurs in the Colombian case, and thus it is possible to estimate a long-term relationship between them. The estimation of the cointegrating vector will be done through the dynamic least squares method in its panel version (See Table 5) [68]. The estimated cointegration vector implies that increases of 1% in dam level are associated with reductions of 0.453% in energy price, and increases of 1% in per capita consumption imply increases of 2.829% in energy price.

**Table 5.** Cointegrated panel.

| Variables | Ln (Exchange Energy Price) |
|---|---|
| - | - |
| Ln (Hydric reserve) | −0.453 ** |
| - | (0.178) [1] |
| Ln (Per capita energy consumption) | 2.829 *** |
| - | (0.287) |
| Observations | 405 |

Notes: [1] Standard errors in parentheses. *** p<0.01, ** p<0.05.

## 5. Discussion and Conclusions

Energy price determination on the Colombia and Norway Energy Exchanges is mainly based on the hydric reserve of the generators. This scenario has fostered proposals to increase costs for water utilization with the main aim of raising awareness for efficient consumption.

This strong influence of water inventory on energy prices means that if dam water levels are high, exchange prices are lower, and when the hydric reserve is low, energy prices rise. In face of this situation, it was found that El Niño phenomenon and drought seasons in the case of Colombia cause the greatest temporary impact on energy prices, since the level of the dams is diminished under such conditions. In the case of Norway, it could be evidenced that increases in hydric reserve are associated with a significant reduction in energy price in said country. For example, in Northern Europe, the last hot and dry summers showed the high vulnerability of the energy sector because of the lower availability of water; hydroelectric generation was reduced with increases in electricity prices because of the high consumption of hydroelectric power [69] [ Considering the integrated information on both countries, it is concluded that increases in dam level are associated with significant reductions in energy price.

Despite the importance of hydric reserve, and of the amount of minute-to-minute information the generators and regulators obtained on this variable, there is no available and public information related to its cost. In a strict sense, this financial information should exist, which would ostensibly improve the quality of Financial Statements since, according to IAS 2, energy must be inventoried through the cost of water use, that is, those costs incurred in transforming water into energy and giving energy its current condition and location. In this way, it is fundamental and of high priority to adequately value the hydric reserve through Financial Accounting with IFRS and Management Accounting with ABC, which are methodologies that respond to good practices implemented worldwide.

Currently, the IFRS are employed by more than 100 countries, and they are standards in terms of financial information according to the New International Financial Architecture, and include measurement, recognition, reporting, and disclosing of information related to transactions and economic events about which it is important to inform the different users. In turn, ABC is one of the most used cost calculation methodologies by service firms worldwide.

Having an adequate methodology that allows to calculate the cost of the hydric reserve not only allows a fundamental basis for energy price determination, but also generates public policy to advance projects of conservation and environmental impact mitigation in hydric basins; this has lead environmental activists to propose tariffs for water use with the main aim of raising awareness toward efficient consumption by all actors; an ideal water use tariff would be one that efficiently allocates the cost to the hydric resource and allows collection of the resources required by environmental management. Additionally, said cost would allow a better classification and presentation of the Financial Statements of hydraulic energy generating companies, enabling in this way decision-making by stakeholders.

It is recommended to conduct future research in topics related to water cost calculation from the accounting point of view through the suggested models which are of international acceptance and propose a water utilization rate close to its real cost. Likewise, it is recommended to deepen research on financial information quality of the financial reports of the hydraulic energy generating companies in the world.

Future research can also address hydraulic energy costs and their impact on sustainability, given that this type of renewable technologies benefit the environment since they generate less emissions and are cost-competitive with respect to generation with non-renewable sources [70]. Among those, hydric is the least costly and most used, which, combined with other renewable sources could contribute to reducing energy price or lead companies to obtain greater profits, according to the economic context of each case [71], because studies have demonstrated that in the short term, using renewable energy generation sources can diminish variability in prices since they would be part of the basis of the generation mix and not as a substitute for balancing supply and demand, making the market less risky [72].

**Author Contributions:** The manuscript was written through join contributions from all authors. The three authors participated in designing the article, defining the theme and methodology, and writing the paper. In this study J.-A.O.-A. obtained the data-source from Norway and Colombia and D.N.-G. and V.R.-F. revised the final version of the paper. All authors have given approval to the final version of the manuscript.

**Funding:** The authors are grateful for the financial support of the Accounting Science Department and Research and Consulting Center (CIC), Research and consulting group in Accounting Sciences (GICCO), and the Research Group in Applied Macroeconomics of the Faculty of Economic Sciences of Universidad de Antioquia. This research received no external funding.

**Conflicts of Interest:** The authors declare no conflict of interest.

## Appendix A

**Table A1.** Sources of electricity production (% of total) and generation of hydroelectric energy (GWh), 2015.

| Country | Hydropower | Coal | Natural Gas | Oil | Renewable Sources | Nuclear Power | Access to Electricity | Generation of Hydroelectric Energy (GWh) |
|---------|-----------|------|-------------|-----|-------------------|---------------|-----------------------|------------------------------------------|
| Norway | 95.8 | 0.1 | 1.8 | 0.0 | 1.9 | 0.0 | 100 | 138,450.0 |
| Paraguay | 100.0 | 0.0 | 0.0 | 0.0 | 0.0 | 0.0 | 99.3 | 55,742.8 |
| Canada | 56.7 | 9.8 | 10.0 | 1.2 | 6.3 | 16.5 | 100 | 382,293.0 |
| Brazil | 61.8 | 4.7 | 13.7 | 5.0 | 12.1 | 2.6 | 99.7 | 359,742.8 |
| Venezuela | 63.7 | 0.0 | 19.4 | 16.9 | 0.0 | 0.0 | 99.5 | 76,003.5 |
| Colombia | 65.0 | 11.9 | 19.3 | 0.6 | 3.3 | 0.0 | 98.2 | 44,681.9 |
| Austria | 60.0 | 8.2 | 12.6 | 1.4 | 16.5 | 0.0 | 100 | 40,592.0 |
| Switzerland | 57.9 | 0.0 | 1.0 | 0.1 | 4.3 | 35.0 | 100 | 39,881.0 |
| Sweden | 46.5 | 0.7 | 0.3 | 0.2 | 16.8 | 34.8 | 100 | 75,439.0 |
| Vietnam | 36.6 | 29.6 | 33.2 | 0.5 | 0.1 | 0.0 | 100 | 58,349.0 |
| China | 19.1 | 70.3 | 2.5 | 0.2 | 4.9 | 2.3 | 100 | 1,130,270.0 |
| Turkey | 25.6 | 29.1 | 37.9 | 0.8 | 6.3 | 0.0 | 100 | 67,146.0 |
| Russia | 15.8 | 14.8 | 49.7 | 0.9 | 0.1 | 17.0 | 100 | 169,920.7 |
| Italy | 16.2 | 16.1 | 39.4 | 4.8 | 22.5 | 0.0 | 100 | 46,970.0 |
| United States | 5.8 | 34.2 | 31.9 | 0.9 | 7.4 | 19.3 | 100 | 271,129.0 |

**Table A2.** Top 20 countries by unutilized hydropower potential.

| Code | Country | Undeveloped (GWh/Year) | Total Potential (GWh/Year) |
|------|---------|------------------------|----------------------------|
| 1 | Russian Federation | 1,509,829 | 1,670,000 |
| 2 | China | 1,013,600 | 2,140,000 |
| 3 | Canada | 805,111 | 1,180,737 |
| 4 | India | 540,000 | 660,000 |
| 5 | Brazil | 435,542 | 817,600 |
| 6 | Indonesia | 388,289 | 401,646 |
| 7 | Peru | 369,058 | 395,118 |
| 8 | DR Congo | 306,512 | 395,118 |
| 9 | Tajikistan | 299,269 | 317,000 |
| 10 | USA | 278,775 | 528,923 |
| 11 | Nepal | 205,777 | 209,338 |
| 12 | Venezuela | 181,163 | 260,720 |
| 13 | Pakistan | 172,820 | 204,000 |
| 14 | Norway | 161,000 | 300,000 |
| 15 | Turkey | 149,100 | 216,000 |
| 16 | Colombia | 151,000 | 200,000 |
| 17 | Angola | 147,048 | 150,000 |
| 18 | Chile | 137,428 | 162,000 |
| 19 | Myanmar | 134,224 | 140,000 |
| 20 | Bolivia | 123,663 | 126,000 |

**Table A3.** Unit root test (ADF).

| Variable | Test Statistic | Critical Value 1% | Critical Value 5% | Critical Value 10% |
|---|---|---|---|---|
| Ln (Exchange Energy Price) | −3.061 | −4.010 | −3.438 | −3.138 |
| Ln (Hydric reserve) | −5.675 | −2.348 | −1.654 | −1.286 |
| Ln (Per capita energy consumption) | −2.304 | −4.01 | −3.438 | −3.138 |

**Table A4.** Bound test.

| Test | Test Statistic | 1% Critical Value | | 5% Critical Value | | 10% Critical Value | |
|---|---|---|---|---|---|---|---|
| F-Test | 6.571 | 5.150 | 6.360 | 3.790 | 4.850 | 3.170 | 4.140 |
| t-Test | −4.353 | −3.430 | −4.100 | −2.860 | −3.530 | −2.570 | −3.210 |

**Table A5.** Error correction model.

| Variables | Adjustment (ADJ) | Long Run (LR) | Short Run (SR) |
|---|---|---|---|
| - | - | | |
| Ln (Exchange Energy Price) = L, | | | 0.178 ** |
| | | | (0.073) [1] |
| Ln (Exchange Energy Price) = L2, | | | −0.160 ** |
| | | | (0.075) |
| Ln (Hydric reserve) = D, | | | −3.034 *** |
| | | | (0.326) |
| Ln (Hydric reserve) = L, | | | 2.362 *** |
| | | | (0.459) |
| Ln (Hydric reserve) = L2, | | | −1.907 *** |
| | | | (0.453) |
| Ln (Hydric reserve) = L3, | | | 1.020 ** |
| | | | (0.400) |
| Ln (Hydric reserve) = L4, | | | −0.408 |
| | | | (0.353) |
| Ln (Hydric reserve) = L5, | | | −0.079 |
| | | | (0.347) |
| Ln (Hydric reserve) = L6, | | | −0.617 * |
| | | | (0.342) |
| Ln (Hydric reserve) = L7, | | | 0.077 |
| | | | (0.347) |
| Ln (Hydric reserve) = L8, | | | −0.390 |
| | | | (0.347) |
| Ln (Hydric reserve) = L9, | | | −0.079 |
| | | | (0.347) |
| Ln (Hydric reserve) = L10, | | | −0.518 |
| | | | (0.337) |
| Ln (Hydric reserve) = L11, | | | 0.950 *** |
| | | | (0.318) |
| Ln (Per capita energy consumption) = D, | | | 0.160 |
| | | | (0.837) |
| Ln (Per capita energy consumption) = L, | | | −0.802 |
| | | | (0.846) |
| Ln (Per capita energy consumption) = L2, | | | −2.307 *** |
| | | | (0.840) |
| El Niño | | | 0.197 ** |
| | | | (0.088) |
| Ln (Hydric reserve) = L, | | −0.155 | |
| | | (0.947) | |
| Ln (Per capita energy consumption) = L, | | 2.389 *** | |
| | | (0.413) | |
| Ln (Exchange Energy Price) = L, | −0.210 *** | | |
| | (0.048) | | |
| Constant | | | 0.068 |
| | | | (3.230) |
| Observations | 192 | 192 | 192 |
| R-squared | 0.510 | 0.510 | 0.510 |

Notes: [1] Standard errors in parentheses. *** p<0.01, ** p<0.05, * p<0.1

**Table A6.** Unit root test on residuals (ADF).

| Variable | Test Statistic | Critical Value 1% | Critical Value 5% | Critical Value 10% |
|---|---|---|---|---|
| Residuals | −3.174 | −2.589 | −1.95 | −1.615 |

**Table A7.** Portmanteau test.

| Test | Test Statistic | P-Value |
|---|---|---|
| Q Statistic (1) | 0.210 | 0.647 |
| Q Statistic (6) | 4.845 | 0.564 |
| Q Statistic (12) | 7.226 | 0.8423 |

**Table A8.** Unit root test (ADF).

| Variable | Test Statistic | Critical Value 1% | Critical Value 5% | Critical Value 10% |
|---|---|---|---|---|
| Ln(Exchange Energy Price) | −3.125 | −2.348 | −1.654 | −1.286 |
| Ln (Hydric reserve) | −3.992 | −2.348 | −1.654 | −1.286 |
| Ln (Per capita energy consumption) | −2.773 | −4.01 | −3.438 | −3.138 |

**Table A9.** Bound test.

| Test | Test Statistic | Critical Value 1% | | Critical Value 5% | | Critical Value 10% | |
|---|---|---|---|---|---|---|---|
| F-Test | 4.797 | 5.150 | 6.360 | 3.790 | 4.850 | 3.170 | 4.140 |
| t-Test | −3.158 | −3.430 | −4.100 | −2.860 | −3.530 | −2.570 | −3.210 |

**Table A10.** Panel unit root test.

| Variable | LLC | Breitung |
|---|---|---|
| Ln (Exchange Energy Price) | 0.004 | 0.100 |
| Ln (Hydric reserve) | 1.000 | 0.000 |
| Ln (Per capita energy consumption) | 0.661 | 0.943 |

**Table A11.** Panel cointegration test.

| Statistic | Value | Z-value | Robust P-value |
|---|---|---|---|
| Pt | −3.947 | −1.973 | 0.015 |
| Pa | −17.063 | −4.202 | 0.010 |

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
