# Peer review of "Cost of Water Use for Negotiating Rates in Energy Exchanges: Evidence from the Hydroelectric Industry"

_water, doi:10.3390/w12020361_

Round 1

Reviewer 1 Report

The paper “Cost of Water Use for Negotiating Rates in Energy Exchanges: Evidence from the Hydroelectric Industry” analyses the importance of the cost of dam water use in hydroelectric generators according to the International Financial Reporting Standard (IFRS) and Management Accounting. I think this paper should be re-written and submitted again to a Journal in a new reviewing process. Maybe Authors should consider submitting this manuscript to a journal more specific I statistics… I definitely do not consider appropriate for publication in water. So, I reject this for publication because of the following reasons:

I think Abstract should be better described. In my opinion, authors are performing a data modelling case study. And, as many researchers are performing data modelling, more sophisticated software is advised… My point is, how do we know these variables are the right ones? How many variables were selected? How do they select a model for Norway, another for Colombia, and a new model considering both data? Why? Which is the best option? Is it possible that any parameter was raised by 2 (or by -5)? This should be considered… Moreover, I would expect to see that the simulations performed resulted in a good system or not… In my opinion, section 2.2 and section 2.3 should be removed Section 3 Material and Methods is the key section to describe the method to obtain results. In my opinion, this section should be better described. As commented before, the reader expects to get final results and not only with simple linear regression.

Author Response

Dear Reviewer 1,

Thanks for your valuable comments. Below you will find our responses to each of your suggestions.

1. I think Abstract should be better described

Response: This suggestion was made by two reviewers in the first review and the improvement suggested by both was taken into account, so the abstract was reorganized to evidence and highlight the scientific importance of accounting, especially that of the International Financial Reporting Standards (IFRS) and Management Accounting, in valuing the hydric reserve.

In the second review, another reviewer expressed that the abstract was well described  and thus suggested making some changes in the introduction to fit the abstract. Therefore, in this case the improvements are made in the introduction following such suggestions.

2. In my opinion, authors are performing a data modelling case study. And, as many researchers are performing data modelling, more sophisticated software is advised… My point is, how do we know these variables are the right ones? How many variables were selected? How do they select a model for Norway, another for Colombia, and a new model considering both data? Why? Which is the best option? Is it possible that any parameter was raised by 2 (or by -5)? This should be considered

Response:  Following the first review suggestion, the econometric model was completely modified to give greater rigor and reliability to the results. Therefore, the correlation between energy price and hydric reserve is estimated in the form of a long-term relationship or level relationship, recognizing the short-term dynamics implicit in this type of relationships. Since time series data intervene in the estimation, unit root tests are conducted on the data to verify their seasonality, applying the Augmented Dickey-Fuller test (ADF).

The mix of seasonal and non-seasonal variables implies that the long-term relationship between the variables cannot be estimated via OLS, nor can the presence of integration between them be directly verified. For the above, the Autoregressive Distributed Lag (ARDL) model methodology is used, verifying whether or not there is a long-term relationship between the variables through the Pesaran, Shin and Smith (PSS) Bound Test.

For the Colombian variables, we work with the ARDL model (3,12,3), that is, a model including three lags of the dependent variable, twelve lags of the hydric reserve logarithm and three lags of the per capita energy consumption logarithm. With this parameterization, and including the constant, the PSS test is conducted. For the Norwegian case, we conclude that there is no long-term relationship and the regressors are zero-order integrated. Given this, the level relationship we seek to obtain can be estimated via OLS, taking the precaution of carrying out the estimation with robust errors due to the presence of autocorrelation so that the inference be reliable. In this sense, the estimation is performed via OLS with robust errors using the Newey-West estimator.

3. Moreover, I would expect to see that the simulations performed resulted in a good system or not… In my opinion, section 2.2 and section 2.3 should be removed Section 3 Material and Methods is the key section to describe the method to obtain results. In my opinion, this section should be better described. As commented before, the reader expects to get final results and not only with simple linear regression.

Response: Following the suggestions by some of the reviewers in the first and second reviews, sections 2.2, 2.3 and section 3 were almost entirely rewritten and improved, for a better understanding from the reader.

Reviewer 2 Report

The paper may be accepted in the present form.

Author Response

Dear Reviewer 2,

Thanks for your valuable comments. Below you will find our responses to each of your suggestions.

1. The paper may be accepted in the present form.

Response: Thanks for the comment, this will encourage us to continue developing our paper.

We hope to have responded satisfactorily to all the suggestions.

Cordially,

Jair-Albeiro Osorio-Agudelo

David Naranjo-Gil

Vicente Ripoll-Feliu

Reviewer 3 Report

The publication seems be interesting for a narrow group of scientists, however it is of good scientific value.

Author Response

Dear Reviewer 3,

Thanks for your valuable comments. Below you will find our responses to each of your suggestions.

1. The publication seems be interesting for a narrow group of scientists, however it is of good scientific value

Response: Thanks for your opinion, we are sure that costing water use for hydraulic energy generation using IFRS and ABC is in fact a contribution to the scientific community.

We hope to have responded satisfactorily to all the suggestions.

Cordially,

Jair-Albeiro Osorio-Agudelo

David Naranjo-Gil

Vicente Ripoll-Feliu

Reviewer 4 Report

The authors thought this job clearly and did a good work of data processing. However, there are still some questions exist in the manuscript, which need to be carefully revised and analysed. I advise major revision.

In the introduction, the authors have emphasised its analysis in reference [4] and it may be excessive and the writing is not sufficiently clearly.

The objective of this work is better defined in abstract than manuscript. Please, the authors should rewrite the third and fourth paragraphs in Introduction where the objective is explained.

A suggestion, the expression “econometric estimations are performed to model the variation in energy price on the Energy Exchange and a variable indicative of the cost of dam water use for both countries” should be before, join to the objective, to explain the importance of this work.

In the 2.1 part, there is a hydroelectric plants classification very simplified. Besides, dams have environmental benefits recognized as flood regulation.

The format of expressions (1) and (2) is excessive. Each expression should be written with abbreviations (i.e. Cost of Energy Generation is CEG) followed by the definitions of the different components of each of them.

In addition, equation (1) and (2) define the same concept Cost of Energy Generation. The expression (2) is a extension or more detailed explanation of the (1)? This part is confused.

In material and methods, the authors have considered 5 variables and they have studied the correlation between energy price and hydric reserve. This part, including the error study, and their conclusions is sufficiently clearly.

However, in which part of the study is the cost study detailed in part 2.1 applied?

what is your relationship with International Financial Reporting Standard (IFRS) and Management Accounting?

This second question is the key question to understand and value this work

Author Response

Dear Reviewer 4,

Thanks for your valuable comments. Below you will find our responses to each of your suggestions.

1. In the introduction, the authors have emphasised its analysis in reference [4] and it may be excessive and the writing is not sufficiently clearly.

Response: The reviewer’s suggestion is followed. Thus, the writing related to such reference is reorganized and improved, highlighting the aspects that contribute to the work, seeking greater clarity and understanding to the reader.

2. The objective of this work is better defined in abstract than manuscript. Please, the authors should rewrite the third and fourth paragraphs in Introduction where the objective is explained.

Response: The reviewer’s suggestion is followed and the introduction paragraphs are rewritten in which the objective is explained, in coherence with the abstract and improving the definition of the objective.

3. A suggestion, the expression “econometric estimations are performed to model the variation in energy price on the Energy Exchange and a variable indicative of the cost of dam water use for both countries” should be before, join to the objective, to explain the importance of this work.

Response: Following the reviewer’s suggestions, the paragraph is rewritten, adding it to the section that makes reference to the objective.

4. In the 2.1 part, there is a hydroelectric plants classification very simplified. Besides, dams have environmental benefits recognized as flood regulation. 

Response: The reviewer’s suggestion is followed by providing more details on the types of hydroelectric plants classifications; the writing is modified taking into account that there are multiple plants classifications, and it is clarified that the Paper uses a simplified classification from the technical perspective according to: Sanz, J. Energías Renovables. Energía Hidroeléctrica, 2a edición.; Prensas Universitarias de Zaragoza, Ed.; ISBN: 9788416933310; 2017.

Additionally, the text is complemented with the benefits or uses this type of plants can have; and given the context of the paragraph, the reference to third party costs is eliminated to fit the new structure of the paragraph.

5. The format of expressions (1) and (2) is excessive. Each expression should be written with abbreviations (i.e. Cost of Energy Generation is CEG) followed by the definitions of the different components of each of them.

Response: The reviewer’s suggestion is followed by modifying the format of the expressions to a more abbreviated one, followed by the meaning of each acronym.

6. In addition, equation (1) and (2) define the same concept Cost of Energy Generation. The expression (2) is a extension or more detailed explanation of the (1)? This part is confused.

Response: The reviewer’s suggestion is followed and one of the equations is eliminated to be more concise and avoid confusion, opting to keep the equation that better expresses the proposal of the paper.

7.In material and methods, the authors have considered 5 variables and they have studied the correlation between energy price and hydric reserve. This part, including the error study, and their conclusions is sufficiently clearly. However, in which part of the study is the cost study detailed in part 2.1 applied?

Response: The reviewer’s comment is totally valid; however, it was not possible to apply the study cost detailed in section 2.1. given that the costs of hydroelectric companies are not disclosed in such detail, nor are they available in the financial statements or in the explanatory notes. Therefore, we resorted to the price and hydric reserve variables to apply the econometric model, given that these implicitly contain the cost of water and hence bring us closer to the expected results.

8. What is your relationship with International Financial Reporting Standard (IFRS) and Management Accounting? This second question is the key question to understand and value this work

Response: Accounting is the social science in charge of studying, measuring, analyzing and keeping a record of the equity of organizations, companies and individuals, with the aim of helping in decision making and control. It is divided mainly in two branches: Financial Accounting, which is compulsory and regulated through the IFRS employed by users external to the organization; and the so-called Management Accounting, which is employed by internal users and is not regulated or compulsory; both are mutually complemented, in particular because of the cost-related information.

We, the authors, are university professors in subjects related to costs for both Financial accounting and Management Accounting purposes. In this sense we have conducted applied cost-related research under the IFRS regulations, such as costs for strictly managerial purposes in multiple economic sectors, in particular the Hydroelectric Sector, the Port Sector, the University Sector and the Industrial Sector.

We hope to have responded satisfactorily to all the suggestions.

Cordially,

Jair-Albeiro Osorio-Agudelo

David Naranjo-Gil

Vicente Ripoll-Feliu

Reviewer 5 Report

Change Roman Numberals to Arabic numbers in first paragraph of intro. For first paragraph of section 2.2 - Identify the market before outlining the electricity production. The electricity production sentence is meaningless until you know where it is. Also, if possible, present all of 2018 - not just June. One month tends to have different characteristics than a whole year. Have a subsection for the different locations. Also for section 2.2 - installed capacity and electricity production are related - but distinct. Pick one and go with it for both locations.  Second to last paragraph in section 2.2 is confusing. Please revise.  Last paragraph of Section 2.2 - does it work? Is the law followed? Please tell me why I care about this law and how it relates to this whole paper.  Paragraph 2 in Section 3 - do you have the historical data for the market prices, hydro availability and customer load? If so, it would be helpful to clarify that. Also, what is the resolution of the data? Is it averaged by month? Is it hourly? Is it all consistent? Where there changes in the percentage of energy served by hydro during the 2000-2016 time period? If so, where those accounted for in some way? How did the prices change over this period? Is hydro availability the strongest indicator?  Discussion - Add one additional paragraph about the integration of renewable energy and how this may change the value of hydropower energy.  Acknowledgement - One "y" needs to be changed to "and". 

Author Response

Dear Reviewer 4,

Thanks for your valuable comments. Below you will find our responses to each of your suggestions.

1. Change Roman Numberals to Arabic numbers in first paragraph of intro.

Response: The reviewer’s suggestion is followed and Roman numbers are changed to Arabic numbers.

2. For first paragraph of section 2.2 - Identify the market before outlining the electricity production. The electricity production sentence is meaningless until you know where it is.

Response: The reviewer’s suggestion is followed by specifying the market for which information of electricity production percentage is provided.

3. Also, if possible, present all of 2018 - not just June. One month tends to have different characteristics than a whole year.

Response The reviewer’s suggestion is followed and an updated source is obtained where the consolidated information on Norway’s energy market for 2018 is presented; the same is done for Colombia.

4. Have a subsection for the different locations.

Response: In the first review it was suggested to join both locations, which were initially separated under independent section numbers but in a consecutive way so, following the suggestion, both locations were joined in single section. However, more clarity is given so that it is understood which country the information provided belongs to.

5. Also for section 2.2 - installed capacity and electricity production are related - but distinct. Pick one and go with it for both locations. 

Response: The reviewer’s suggestion is followed, and we opt to keep the concept of electricity production for both countries, so that for Colombia it is replaced by the electric energy production for 2018 in the section that mentioned installed capacity.

6. Second to last paragraph in section 2.2 is confusing. Please revise. 

Response: Following other suggestions made regarding this chapter, we made changes in all the section and, with this, expect to have eliminated any confusing ideas.

7. Last paragraph of Section 2.2 - does it work? Is the law followed? Please tell me why I care about this law and how it relates to this whole paper. 

Response: Article 43 of Colombian Law 99 of 1993 is still in effect. It refers to a tax that is charged to anyone requiring environmental licenses involving the use of water taken from natural sources, so it is similar to what would be a cost for water use, and in this case the money is allocated for the recovery, preservation, conservation and surveillance of the hydrographic basin feeding the respective hydric source. However, according to the reviewer’s argument, we realize the interpretation of the idea in the paragraph can lead to confusion with respect to the proposal in the Paper, so we decided to eliminate it, following the suggestion.

8. Paragraph 2 in Section 3 - do you have the historical data for the market prices, hydro availability and customer load? If so, it would be helpful to clarify that. Also, what is the resolution of the data? Is it averaged by month? Is it hourly? Is it all consistent?

Response: The reviewer’s suggestion is followed, the paragraph is rewritten and the corresponding clarifications are made regarding the different variables for a better understanding from the reader.

9. Where there changes in the percentage of energy served by hydro during the 2000-2016 time period? If so, where those accounted for in some way? How did the prices change over this period? Is hydro availability the strongest indicator? 

Response: According to the behavior of the hydric reserve variable, it does not indicate this type of events; thus, price changes are the ones capturing the model through the estimated parameter. We believe that the indicator of hydroelectric availability in the country is stronger, given the importance of this type of electric generation. Since there are no drastic changes in the reserve level, the effects of its changes are captured through the model.

10. Discussion - Add one additional paragraph about the integration of renewable energy and how this may change the value of hydropower energy.  Acknowledgement - One "y" needs to be changed to "and". 

Response: The reviewer’s suggestion is followed, so a paragraph is added at the end of the discussion, making reference to the changes in hydroelectric energy price with the integration of renewable energies.

We hope to have responded satisfactorily to all the suggestions.

Cordially,

Jair-Albeiro Osorio-Agudelo

David Naranjo-Gil

Vicente Ripoll-Feliu

Round 2

Reviewer 1 Report

The paper “Cost of Water Use for Negotiating Rates in Energy Exchanges: Evidence from the Hydroelectric Industry” analyses the importance of the cost of dam water use in hydroelectric generators according to the International Financial Reporting Standard (IFRS) and Management Accounting.

I think authors have made the corrections successfully and the level of the manuscript has increased after reviewing process. After these changes, this paper has been improved and now it deserves publication in water. 

Reviewer 4 Report

I Think authors have carefully and significantly revised the manuscript according to all reviewers' comments, with suggestions were appropriately addressed. However, for future reviews, it is difficult for the reviewers to check the revisions because the changes are marked in the manuscript but not the previous version or why this paragraph has been modified. I suggest this paper for publication.

This manuscript is a resubmission of an earlier submission. The following is a list of the peer review reports and author responses from that submission.

Round 1

Reviewer 1 Report

The paper “Cost of Water Use for Negotiating Rates in Energy Exchanges: Evidence from the Hydroelectric Industry” helps to calculate the exchange price pf energy through several parameters (hydric reserve), per capita energy consumption. This manuscript is well written in English but I think this paper does not deserve to continue in this reviewing process and I reject this for publication because of the following reasons:

1.      Abstract is a bit weak. Authors are only performing a data modelling study.

2.      The paper is not well structured. The literature review process shows interesting information but it does not belong (in my opinion) to a scientific document. I would send this information to an Appendix. This section seems to be taken from a PhD. As it shows a lot of information about the Norwegian and Colombian case.  Figure 1-7should be removed.

3.      Only the paragraph between Figure 2 and Figure 3. From “The electricity market…. “and up to “the high residential demand” is relevant for this case in the Norwegian case.

4.      Table 2 should clearly show which costs are investment and/or maintenance cost to be used in the cost analysis.

5.      I would like to highlight that section 2 is really hard to read and should be removed. 11 pages and nothing is shown to the reader.

6.      Section 3 Material and Methods: the method to obtain results is quite weak. The reader expect to get final results with more powerful methodologies than simple linear regression as they expect to find the better relationship that was useful to predict the Exchange price.

7.      The methodology is not well explained. The results should be shown (expecting better fitting to data with powerful methodologies).

8.      Table 4, 5 and 6 should be in a single Table

9.      Conclusions should also be improved.

Reviewer 2 Report

Referee report for “Cost of Water Use for Negotiating Rates in Energy Exchanges: Evidence from the Hydroelectric Industry”

The article claims to analyze the importance of the cost of water use in hydroelectric generators using established international accounting standards. It also studies the correlation between the hydric reserves in river dams and the energy price set in energy markets through the case study of Norway and Colombia. It finds a negative correlation between the exchange price of energy and the level of hydric reserves in river dams. This subject falls within the aims and scope of the journal.

I believe the paper should be rejected because the authors claim that a mere proposal is a significant contribution and the empirical results they find are neither reliable nor particularly relevant.

A high importance is given in the abstract and the conclusion to the proposal of applying international accounting standards (IFRS) to the calculation of the cost of hydropower production, but it is a simple nonscientific proposal and definitely not a finding or a new accounting method being developed.

I have serious doubts about the soundness of the empirical study presented (described below in the major comments) and the conclusions (energy price is positively associated with energy consumption and scarcity of the based resource) are not particularly relevant to merit publication in an international peer-reviewed journal.

Below I list some major and minor issues to be addressed by the authors before future submissions.

Major issues

Page 1, abstract: The following finding is claimed in the abstract “there is a positive and significant relationship between the cost of water and the price of energy”. Although the statement is something to be expected, the paper provides no evidence of this. The cost of water is neither presented nor calculated in the paper.

P. 2, introduction, last paragraph. The choice of Norway and Colombia alone calls for a better justification. They are not alone in the group of countries with a high importance of hydropower for electricity production and established energy exchange markets. Working with a panel of countries would provide more reliable results.

P. 2-3, literature review. The first 5 paragraphs (out of 10) of the literature review section are proper for an introduction, not for a literature review. The following paragraphs refer to some papers or books in which the components of the cost of water use are described or calculated, namely the opportunity costs of water use. However, the references have no bearing on the work presented by the authors as no attempt is made to calculate neither the financial nor the economic (opportunity) cost of water use for hydropower.

P. 8, section 2.2, last paragraph. “during the period 2011-2017 […], it is observed that Colombia industrial prices are much higher than Norway […], which, according to the information above, is not due to water cost but to other factors”. Something should be said about the relevant factors affecting the energy prices. These prices will constitute the dependent variables in the paper’s regressions and if there are relevant factors impacting them which are not considered in the regressions, the coefficients estimated may be biased.

P. 11. Table 3 presents a list of six cost generating processes of hydropower generation. However, equation 2 shows the total cost as being the sum of only three. The author’s justification is merely that the these three “standout”. This is not rigorous and it is arguably an argument to neglect part of the costs when presenting a supposed method to calculate the total cost.

p. 12, paragraph 5. Taking logs does not avoid the problems of heteroscedasticity. At most, it lessens them in case the variance of the estimator is proportional to the level of the regressor.

Sections 3 and 4. The use of OLS on time series without proper care whether the OLS assumptions hold is wrong and leads to biased results. I see neither tests for serial correlation of the observations nor the potential introduction of lagged variables to deal with it. Given the nature of the variables and the structure of the data used (time series), the presence of serial correlation is something I would expect. The consequences may be serious in terms of bias in the coefficients estimated. The fact that table 5 presents a negative coefficient (although not significant) for per capita energy consumption is strange and should have alerted the authors to implement post-estimation tests to the soundness of their regression results. Furthermore, I see no reason for seasonal variables to be omitted from the estimation for Colombia and for the combined case. Given their significance for the case of Norway, not having them in the combined case may introduce omitted variable bias. Also, summary statistics should have been presented for the variables used.

Minor issues

P. 3, 3rd paragraph, line 1: “Dams technically have a way to use water”. Obviously, but given that nothing else is said about it, this kind of sentence is useless.

P. 3, 3rd paragraph, line 5: “implementation of this directive has been slow in terms of governance”. No evidence provided. There are public reports available regarding the implementation of the Water Framework Directive.

P. 9, table 2. The expression “during the construction period” is repeated.

P. 12, line 5. Replace “International Energy Agent” for “International Energy Agency”.

P. 12, paragraph 4. Correct “Fort the purpose”

Reviewer 3 Report

The presented paper is interesting and easy to read. The subject brings some novelty into the discussion what is good among the scientific community.

Although this is an empirical study, it brings interesting results that deserve our attention. These results are someway obvious however, this study proves the existence of such relations in a macro scale.

Good grammar and well structured. Congrats for the idea.